# Enhancing Feature Diversity Boosts Channel-Adaptive Vision Transformers

**Chau Pham**
Boston University
Boston, MA
chaupham@bu.edu

**Bryan A. Plummer**
Boston University
Boston, MA
bplum@bu.edu

## Abstract

Multi-Channel Imaging (MCI) contains an array of challenges for encoding useful feature representations not present in traditional images. For example, images from two different satellites may both contain RGB channels, but the remaining channels can be different for each imaging source. Thus, MCI models must support a variety of channel configurations at test time. Recent work has extended traditional visual encoders for MCI, such as Vision Transformers (ViT), by supplementing pixel information with an encoding representing the channel configuration. However, these methods treat each channel equally, *i.e.*, they do not consider the unique properties of each channel type, which can result in needless and potentially harmful redundancies in the learned features. For example, if RGB channels are always present, the other channels can focus on extracting information that cannot be captured by the RGB channels. To this end, we propose $\mathrm{DiChaViT}$, which aims to enhance the diversity in the learned features of MCI-ViT models. This is achieved through a novel channel sampling strategy that encourages the selection of more distinct channel sets for training. Additionally, we employ regularization and initialization techniques to increase the likelihood that new information is learned from each channel. Many of our improvements are architecture agnostic and can be incorporated into new architectures as they are developed. Experiments on both satellite and cell microscopy datasets, CHAMMI, JUMP-CP, and So2Sat, report $\mathrm{DiChaViT}$ yields a $1.5 - 5.0\%$ gain over the state-of-the-art. Our code is publicly available at https://github.com/chaudatascience/diverse_channel_vit.

## 1 Introduction

Most visual encoders assume they are provided with a fixed-channel representation as input (*e.g.*, they take RGB inputs as input at train and test time) [1–10]. However, many applications find a variety of imaging techniques beyond just the traditional RGB channels beneficial. For example, satellite images or sensors onboard a robot often contain an infrared camera in addition to traditional RGB, and microscopes can also host a significant range of potential imaging channels [11–17]. Thus, Multi-Channel Imaging (MCI) models aim to learn good feature representations from datasets with heterogeneous channels, where the number and type of channels can vary for each input at test time. Training a model that is robust to changes in channel configurations can save time and resources as only a single model needs to be learned, while also helping to prevent overfitting in small datasets through transfer learning [14]. Prior work proposed methods to make MCI models robust to missing channels by randomly masking them during training [18]. As shown in Fig. 1(a) left and (b) top, this results in redundancies being learned across channels during training rather than encoding new information. A consequence of this repetition is a model focused on learning strong cues that are easy to identify, making it less capable of learning unique and/or challenging cues within each channel.

38th Conference on Neural Information Processing Systems (NeurIPS 2024).

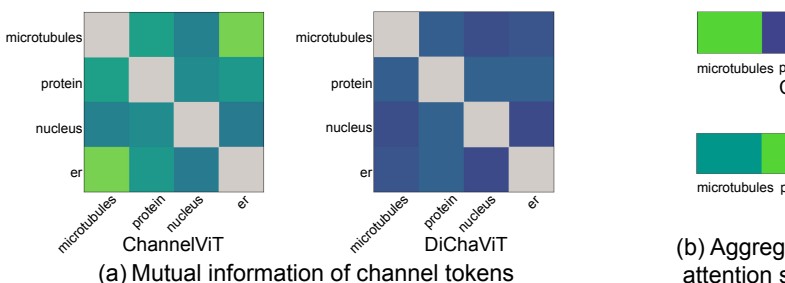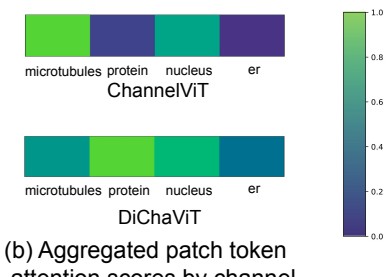

Figure 1: **Comparison of the redundant information learned by different models on the HPA dataset in CHAMMI [14]).** **(a)** Measures the mutual information between the channel tokens, which captured the configuration of channels in an image. Note we gray out the diagonal for better visualization. We find ChannelViT tokens have high mutual information, which suggests significant redundancy exists across channels [34, 35]. In contrast, DiChaViT has little mutual information as each channel is encouraged to learn different features. **(b)** We compute attention scores of the [CLS] token to the patch tokens in the penultimate layers and aggregate them by channel. ChannelViT (top) relies on certain channels (*e.g.*, *microtubules* and *nucleus*) to make predictions and less on other channels (*e.g.*, *protein* and *er*). In contrast, DiChaViT demonstrates more evenly distributed attention scores across channels, suggesting that each channel contributes more to the model's predictions.

To address this limitation, we propose a **Di**verse **Cha**nnel **Vi**sion **T**ransformer (DiChaViT) that aims to balance the robustness in different MCI configurations, which may cause redundancy, with a need to learning diverse and informative features. First, we include a Channel Diversification Loss (CDL), a regularization term that encourages a special channel token, which represents the presence of a channel in the input data, to be distinct from the other channel tokens. As shown in Fig. 1(a) right, this reduces repeated information across our model's channels. However, this can still result in similar features being encoded for each image patch. Thus, our Token Diversification Loss (TDL) aims to directly diversify the features learned for each patch token as shown in the bottom of Fig. 1(b) by encouraging that each patch token is orthogonal to the others. Finally, rather than a uniform random channel masking strategy as used in prior work [18, 19], we introduce Diverse Channel Sampling (DCS), in which we select channels based on their dissimilarity, further promoting feature diversity. We observe that promoting a more diverse representation enables each channel to contribute more to the final prediction, leading to a performance boost of up to 5.0% in downstream MCI tasks. Fig. 2 provides an overview of our approach.

The work that is closest in spirit to ours are methods that are designed to learn disentangled representations [20–27], *e.g.*, learning features aligned to a given set of attributes [28–30]. These methods have shown a trade-off between the strength of the disentanglement and the downstream tasks performance [31–33]. This is due, in part, to the fact that many attributes these methods aim to disentangle are correlated with each other, making it challenging to know what features relate individually to each attribute. However, unlike these tasks, MCI methods do not focus only on disentangling features across channels. Instead, they must capture some redundant information to be robust to missing channels while simultaneously learning features that may only arise in a subset (or even a single) channel. In other words, in MCI some redundancy is desirable across channels even if we could learn perfectly disentangled representations. In addition, many methods in disentangled representation learning assume the attributes to separate are labeled, but there are no labeled attributes in MCI. Instead, DiChaViT must automatically decide what to capture in multiple channels while still learning important channel-specific information.

We summarize our contributions below:

- We propose DiChaViT as a solution to enhance feature diversity and robustness in MCI-ViTs, boosting classification accuracy by $1.5 - 5.0\%$ over the state-of-the-art on three diverse MCI datasets: CHAMMI [14], JUMP-CP [12], and So2Sat [17].
- We introduce a new channel sampling strategy to encourage the selection of more distinct channel sets during training, thereby enhancing feature diversity in MCI models.
- We introduce regularization and initialization techniques that better balance robustness to different configurations in MCI and facilitate learning diverse and informative features.

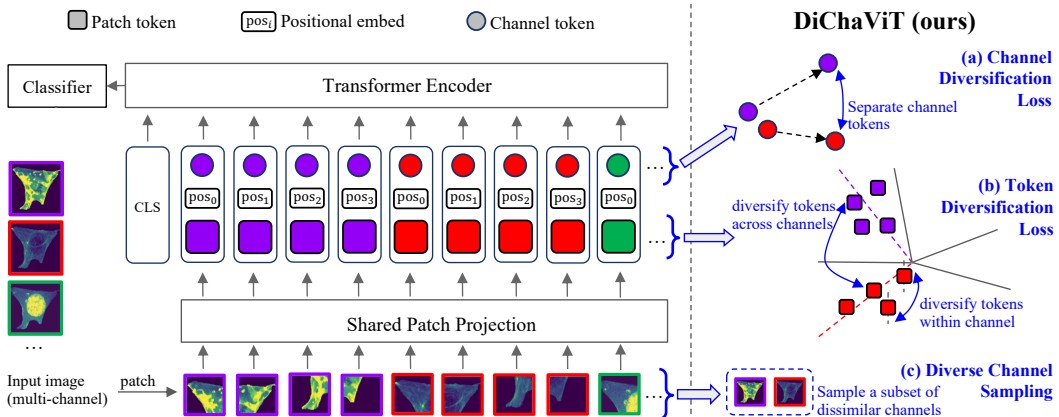

Figure 2: **An overview of DiChaViT.** We introduce two regularization methods on the features and a channel sampling strategy to promote diversity in feature representations. We apply **(a)** Channel Diversification Loss (CDL) (Sec. 3.1) for channel tokens (●), and **(b)** Token Diversification Loss (TDL) (Sec. 3.2) on the patch tokens (■). Additionally, we **(c)** sample a subset of dissimilar channels using Diverse Channel Sampling (DCS) (Sec. 3.3).

## 2 Related Work

**Convolutional-based models for multi-channel imaging.** Researchers have been developing convolutional-based models to keep pace with the evolving landscape of multi-channel imaging data. Bhattacharyya *et al*. [36] introduced $\mathrm{IRFacExNet}$, which utilizes depth-wise convolutions to merge channel-wise features from infrared thermal images. Jiang *et al*. [37] introduced a double-channel CNN that takes into account the correlation between input channels in aerial images. This approach employs a separate sub-network for each group of channels and then performs feature fusion to aggregate features across channels. Siegismund *et al*. [38] presented $\mathrm{DCMIX}$ to work with images with many channels based on imaging blending concepts. While these methods can be used for MCI, they are not designed to work on varying input channels. In a recent study, Chen *et al*. [14] introduced and adapted channel-adaptive models based $\mathrm{Depthwise}$ convolutions, $\mathrm{TemplateMixing}$ [39–41], and $\mathrm{HyperNets}$ [42]. These models incorporate their adaptive interface in the first layer of an otherwise shared $\mathrm{ConvNeXt}$ model [8]. While these methods provide a strong baseline, they find settings where some channels are missing during inference challenging. In our work, we aim to improve MCI model robustness by improving the diversity of learned features.

**Vision Transformers for multi-channel imaging**. Vision transformers (ViT) [43] have natural advantages when dealing with multiple channels, especially when the number of channels varies. ViTs treat image modeling as sequence-to-sequence problems, allowing them to be flexible in handling different numbers of image tokens. Nguyen *et al*. [44] introduced *variable tokenization* and *variable aggregation*, in which they divided each input channel independently into patches and then aggregated the patch features across channels using learnable queries. Tarasiou *et al*. [45] proposed TSViT, which incorporates a tokenization scheme and temporal position encodings to process Satellite Image Time Series. In a relevant work, Zhou *et al*. [46] introduced FAN, a channel reweighting design aimed at adjusting channel features based on the observation that some channels capture more significant information than others. In the medical domain, Hatamizadeh *et al*. [47] proposed UNETR that utilized a transformer encoder followed by a skip-connected decoder for 3-D medical image segmentation. Recently, Bao *et al*. [18] proposed $\mathrm{ChannelViT}$ that processes each input channel independently via a shared linear projection and incorporates a learnable channel embedding for preserving channel-specific features. In addition, the authors proposed Hierarchical Channel Sampling (HCS), a regularization technique applied to the input channels to boost robustness and reduce training time. $\mathrm{ChannelViT}$ outperforms standard ViTs in classification tasks and demonstrates its generalization ability when only a subset of the trained channels is available during inference. In a similar work, Bourriez *et al*. [15] introduced $\mathrm{ChAda\text{-}ViT}$, a channel adaptive attention technique for handling heterogeneous microscope images. However, these methods do not adequately model the unique properties of each channel type, resulting in harmful redundancies, whereas we boost the diversity of features across channels to enhance the robustness of MCI-ViT models.

# 3 Encouraging Diverse Representations in multi-channel ViTs

Given a multi-channel image (MCI) $X$ containing channels $c_i \in C_X$, our goal is to train a model $M$ that takes our input image $X$ as input to make its predictions. Following [15, 18], we consider the MCI setting where $M$ has seen all the channels we expect to see during inference, *i.e.*, $C_X \subseteq C_M$. We leave the exploration of handling novel channels during inference for future work, as it presents significant challenges, including establishing meaningful connections between existing and new channels, and identifying informative channel weights in the presence of domain shifts. In our setting, since we do not know what $C_X$ we may see during inference, prior work has focused primarily on exploring methods that are robust to different choices of $C_X$ by encouraging $M$ to redundancies across channels (*e.g.*, [14, 15, 18]). Specifically, they begin with a base ViT encoder [48] that uses each channel-specific image patch $p_i$ as input. Each image patch is passed through a shared patch projection layer and concatenated with its corresponding channel token $ch_i$. Hierarchical Channel Sampling (HCS) [18] encourages robustness to missing channels by randomly masking some channels during training to ensure key information can be captured in multiple channels. However, as noted in the Introduction, this can be harmful when $M$ does not balance this repetitive feature learning to also capture distinctive channel-specific information.

As illustrated in Fig. 1, DiChaViT aims to better balance repetitive and distinct feature learning through three major components. First, we use a Channel Diversification Loss (CDL) to learn diverse representations to help prevent feature collapse in the channel tokens (Sec. 3.1). Second, our Token Diversification Loss (TDL) encourages patch tokens to also learn distinct features (Sec. 3.2). Finally, Diverse Channel Sampling (DCS) promotes robustness to missing channels while also encouraging that new features are also learned during training (Sec. 3.3). These components enable our approach to balance repetitive and channel-specific feature learning (overview in Fig. 2).

## 3.1 Enhancing channel token separation

Recall that in Fig. 1(a), learned channel tokens $ch_i$ from prior work show high mutual information, indicating these tokens are not well-separated. Following [41, 49–51], we partly mitigate this issue by replacing the random initialization of $ch_i$ used by prior work [15, 18] with an orthogonal initialization. To further encourage the diversity in the features, we introduce Channel Diversification Loss (CDL) for increased separation between the channel tokens (Fig. 2(a)). Inspired by $\mathrm{ProxyNCA}{+}{+}$ [52], the idea is to use a learnable vector (*i.e.*, an orthogonally initialized *channel anchor*) to represent each channel in the input image during training. We promote diversity in the channel tokens by pulling channel features toward their corresponding anchors while pushing them away from all other anchors. A key benefit of this approach is that the anchors prevent channel tokens from collapsing while still allowing for flexibility in learning useful representations.

Formally, we denote $A$ as the set of all channel anchors, $t_{\mathrm{CDL}}$ as the temperature, and $\| \cdot \|_2$ as the $L2$-Norm. We start by initializing the channel tokens $ch_i$ and their channel anchors orthogonally. Then, we apply CDL as follows:

$$\mathcal{L}_{\mathrm{CDL}} = -\log \left( \frac{\exp\left(-d\left(\frac{ch_i}{\|ch_i\|_2}, \frac{g(ch_i)}{\|g(ch_i)\|_2}\right) \cdot \frac{1}{t_{\mathrm{CDL}}}\right)}{\sum_{g(a)\in A} \exp\left(-d\left(\frac{ch_i}{\|ch_i\|_2}, \frac{g(a)}{\|g(a)\|_2}\right) \cdot \frac{1}{t_{\mathrm{CDL}}}\right)} \right), \tag{1}$$

where $g(ch_i)$ is a function that returns a corresponding channel anchor for channel token $ch_i$, and $d(ch_i, g(\cdot))$ is the squared Euclidean distance between channel token $ch_i$ and an anchor. In Eq. 1, the numerator calculates the distance of a channel token to its anchor, while the denominator computes all these distance pairs of the channel token to all the channel anchors. When the temperature value $t_{\mathrm{CDL}}$ is set to 1, we get a standard $\mathrm{Softmax}$ function. Lowering the temperature can lead to a more focused and sharp probability distribution, but we found that the results are not very sensitive to the value of $t_{\mathrm{CDL}}$. Thus, we simply use a fixed temperature $t_{\mathrm{CDL}}$ of $1/14 \approx 0.07$.

## 3.2 Enhancing feature diversity for patch tokens

MCI-ViT models like $\mathrm{ChannelViT}$ [18], $\mathrm{ChAda\text{-}ViT}$ [15] use a shared linear projection to extract features independently from each input channel in the image rather than using separate projections for each channel. With the shared projection, only the common features across channels are retained,

while other channel-specific information is filtered out, which helps to reduce overfitting. However, this design can also produce similar representations for all patch tokens. This is not ideal because each patch may contain unique information that would be ignored. In our approach, we also leverage this shared projection, but we enhance it with Token Diversification Loss (TDL), a regularization applied to the patch token features to enhance the diversity of features learned by each patch in the input image (see Fig. 2(b) for an overview). Specifically, we enforce an orthogonality constraint on the tokens to ensure that each token is orthogonal to the others. Additionally, we take into account the token type information to differentiate between tokens from the same channels and across channels. The main idea is to make features from different channels more distinct while allowing for a certain level of similarity among features within the same channel.

Let $\mathbf{p}_i$ be the input patch at position $i$, and $\mathbf{W}_{\text{proj}}$ be the shared linear projection at the first layer. We denote $\mathbf{t}_i = \mathbf{W}_{\text{proj}} \cdot \mathbf{p}_i$ as the patch feature token of $\mathbf{p}_i$, $T = \{\mathbf{t}_i\}_{i=1,2,\dots}$ as the set containing all patch feature tokens in the input image, and $h(\mathbf{t}_i)$ as a function that returns the corresponding channel for input patch $\mathbf{p}_i$. We devise a unified loss function for each input image as follows:

$$\mathcal{L}_{\text{s}} = \frac{1}{N_s} \sum_{\mathbf{t}_i, \mathbf{t}_j \in T;\ h(\mathbf{t}_i)=h(\mathbf{t}_j)} \langle \mathbf{t}_i, \mathbf{t}_j \rangle \tag{2}$$

$$\mathcal{L}_{\text{d}} = \frac{1}{N_d} \sum_{\mathbf{t}_i, \mathbf{t}_k \in T;\ h(\mathbf{t}_i)\neq h(\mathbf{t}_k)} \langle \mathbf{t}_i, \mathbf{t}_k \rangle \tag{3}$$

$$\mathcal{L}_{\text{TDL}} = \lambda_s \cdot |\mathcal{L}_{\text{s}}| + \lambda_d \cdot |\mathcal{L}_{\text{d}}| \tag{4}$$

where $\langle \cdot, \cdot \rangle$ represents the cosine similarity, $|\cdot|$ denotes an absolute value, and $N_s$, $N_d$ are the numbers of patch token pairs in the two equations respectively. Eq. 2 calculates the average cosine similarity of all feature token pairs in the same channels, while Eq. 3 calculates the average of all feature token pairs from different channels. The two losses are combined with weights $\lambda_s$ and $\lambda_d$ to balance the constraint of tokens belonging to the same channels (first term) and tokens belonging to different channels (second term), to form the final loss $\mathcal{L}_{\text{TDL}}$ in Eq. 4. Our goal is to encourage each patch token to be orthogonal to each other to promote the diversity of patch tokens.

### 3.3 Diverse Channel Sampling (DCS)

Bao *et al.* [18] introduced HCS to reduce the training time and improve the robustness of the model. The main concept is to randomly drop some input channels and train the model only on the remaining channels. In the same spirit, we propose a novel method, Diverse Channel Sampling (DCS), to sample a more diverse subset of channels during training (Fig. 2(c)). Similar to HCS, we start by randomly sampling a number $k$, which is the size of a subset of channels to train on. However, while HCS samples $k$ channels randomly, DCS first samples an anchor channel $c_k$. Then, we select other $k-1$ channels that are dissimilar to the anchor channel. This idea shares similarity with Channel DropBlock [53], where a set of similar channels in a CNN layer is masked out to disrupt co-adapted features. However, instead of keeping a fixed number of feature map channels as in Channel DropBlock, DCS selects a flexible number of input channels for each sampling. The procedure of DCS is outlined in Algorithm 1.

In practice, Algorithm 1 can be applied to a batch of images for faster sampling. We use channel token $\text{ch}_i$ to represent the channel feature $f_i$. Refer to Sec. 4.4 and Tab. 5 for more discussion on choices of $f$. The temperature $t_{\text{DCS}}$ controls the sharpness of the probability distribution. With a large $t_{\text{DCS}}$, DCS reduces to HCS, while with a small $t_{\text{DCS}}$, DCS selects a random subset of channels that are the least similar to the anchor channel.

### 3.4 Training Objective

The final loss consists of the primary loss for the specific task (*e.g.*, cross-entropy for classification), Channel Diversification Loss (CDL) applied to channel tokens, and Token Diversification Loss (TDL) used on patch tokens. These terms work together to promote diversity in channel and patch token features, resulting in a more robust model, as shown in Eq. 5:

$$\mathcal{L}_{\text{final}} = \mathcal{L}_{\text{task}} + \lambda_{\text{CDL}} \cdot \mathcal{L}_{\text{CDL}} + \mathcal{L}_{\text{TDL}} \tag{5}$$

where $\lambda_{\text{CDL}}$ is a weight to balance CDL. Note that TDL is balanced by $\lambda_s$ and $\lambda_d$ in Eq. 4.

---
**Algorithm 1:** Diverse Channel Sampling (DCS)
---
**Input** : Image $X$ with $m$ channels $c_1, ..., c_m$
             Channel feature $f_i$ for each input channel $c_i$
             Temperature $t_{\text{DCS}}$

1 Sample a random variable $k$ uniformly from the set $\{1, 2, ..., m\}$
2 Sample an anchor channel $c_k$ uniformly from all $m$ channels
3 Compute the cosine similarity between channel $c_k$ and the other $m-1$ channels:
    $\mathbf{s} = [\langle f_k, f_i \rangle, ...], \forall i \neq k \quad (\mathbf{s} \in \mathbb{R}^{m-1})$
4 Convert $1 - \mathbf{s}$ to probability using $\text{softmax}$ with temperature $t_{\text{DCS}}$:
    $\mathbf{p} = \text{softmax}((1 - \mathbf{s})/t_{\text{DCS}}) \quad (\mathbf{p} \in \mathbb{R}^{m-1})$
5 Sample $k-1$ distinct channels from $m-1$ channels with probability $\mathbf{p}$
6 Combine the $k-1$ channels with channel $c_k$ to create a set of $k$ sampled channels.
**Output :** Image $X$ with only $k$ sampled channels
---

## 4 Experiments

### 4.1 Experimental Setup

**Baseline methods.** We adopt the following baseline methods.

- **DepthwiseViT** [14] utilizes a depthwise convolution layer to independently filter each input channel. The resulting features are averaged to create a new feature representation, which is then fed into a ViT backbone.
- **TemplateMixingViT** [39, 40] generates weights for each channel by learning a linear combination of shared, learnable parameter templates. These weights are formed into a patch project layer, followed by a ViT backbone.
- **HyperNetViT** [42] employs a neural network (*e.g.*, MLP) to independently generate weights for each channel, which are then concatenated to form a patch projection layer. This patch projection layer is subsequently used in a ViT backbone.
- **ChAda-ViT** [15] uses a shared projection layer to extract features from each channel separately, then feeds these tokens, together with their corresponding positional embeddings and channel embeddings, into a ViT backbone.
- **ChannelViT** [18] is the same general architecture as ChAda-ViT, but also employs Hierarchical Channel Sampling (HCS) during training.

**Implementation details.** As HCS proves robust in multi-channel imaging [18], we incorporate this technique for DepthwiseViT, TemplateMixingViT, and HyperNetViT to ensure a fair comparison in these adaptive baselines used by Chen *et al*. [14][1]. For ChannelViT and ChAda-ViT, due to their similarity (primarily a difference in whether HCS is included), we use the implementation from [18] for both methods[2]. All baselines utilize a ViT small architecture (21M parameters) implemented in DINOv2 [54] as the backbone [3]. We use AdamW optimizer [55] to train the models, minimizing cross-entropy loss on JUMP-CP and So2Sat, and proxy loss on CHAMMI. For the learning rate, we use a scheduler with linear warmup and cosine decay. Refer to Appendix Sec. A for details.

**Metrics.** We evaluated the methods by calculating their top-1 classification accuracy on the So2Sat [17] and JUMP-CP [12] datasets. For CHAMMI [14], we used the evaluation code[4] provided by the authors, in which a 1-Nearest Neighbour classifier is used to predict the macro-average F1-score for each task separately. We report the average score on WTC and HPA, and present the detailed results in Tab. 7 of the Appendix.

### 4.2 Datasets

**CHAMMI [14]** consists of varying-channel images from three sources: WTC-11 hiPSC dataset (WTC-11, three channels), Human Protein Atlas (HPA, four channels), and Cell Painting datasets

---

[1]https://github.com/chaudatascience/channel_adaptive_models
[2]https://github.com/insitro/ChannelViT
[3]https://github.com/facebookresearch/dinov2
[4]https://github.com/broadinstitute/MorphEm

Table 1: **Comparison of test accuracy of channel adaptive models**. "Full" refers to inference on all channels, while "Partial" means testing on a subset of channels (*Sentinel-1* channels for So2Sat, *fluorescence* channels for JUMP-CP). We find our model outperforms other baselines, with a $5.0\%$ boost on CHAMMI and a $1.5 - 2.5\%$ point improvement on JUMP-CP and So2Sat.

| | CHAMMI [14] | JUMP-CP [12] | | So2Sat [17] | |
| --- | --- | --- | --- | --- | --- |
| Model | Avg score | Full | Partial | Full | Partial |
| HyperNetViT [42] | 54.54 | 47.07 | 42.43 | 60.73 | 41.88 |
| DepthwiseViT [14] | 60.94 | 49.86 | 44.98 | 60.41 | 43.41 |
| TemplateMixingViT [39, 40] | 57.02 | 52.48 | 43.85 | 55.86 | 37.28 |
| ChAda-ViT [15] | 63.88 | 65.03 | 42.15 | 56.98 | 12.38 |
| ChannelViT [18] | 64.90 | 67.51 | 56.49 | 61.03 | 46.16 |
| **DiChaViT** (ours) | **69.68** | **69.19** | **57.98** | **63.36** | **47.76** |

Table 2: **Test accuracy of DiChaViT and ChannelViT on partial channels of JUMP-CP [12]**. Each column represents *mean±std* for all combinations when tested on partial channels. For example, column "7" indicates testing on 7 out of 8 channels, and, thus, the reported variance is due to the presence or absence of a channel. See to Tab. 9 in the Appendix for detailed results for each combination for column "7" with model variance. DiChaViT consistently exhibits improved robustness in the presence of missing channels during inference.

| | Number of channels for evaluation | | | | | | | |
| --- | --- | --- | --- | --- | --- | --- | --- | --- |
| Method | 8 | 7 | 6 | 5 | 4 | 3 | 2 | 1 |
| ChannelViT [18] | 67.51 | $60.36_{\pm9.1}$ | $52.74_{\pm12.2}$ | $44.89_{\pm13.2}$ | $36.88_{\pm12.3}$ | $29.36_{\pm9.3}$ | $23.70_{\pm5.0}$ | $20.78_{\pm1.6}$ |
| **DiChaViT** (ours) | 69.19 | $61.91_{\pm9.3}$ | $54.49_{\pm12.4}$ | $46.35_{\pm13.4}$ | $38.00_{\pm12.4}$ | $30.09_{\pm9.3}$ | $23.97_{\pm4.9}$ | $20.90_{\pm1.6}$ |

(CP, five channels). The three sub-datasets contain a total of 220K microscopy images, of which 100K images are for training and the rest for testing across various tasks. The models are trained to learn feature representation and then evaluated on domain generalization tasks.

**JUMP-CP [12]** comprises images and profiles of cells that were individually perturbed using chemical and genetic methods. Our experiments focus on the compound perturbation plate BR00116991, which contains 127K training images, 45K validation images, and 45K test images. Each image has eight channels, with the first five being *fluorescence* and the remaining three containing *brightfield* information. The dataset consists of 161 classes, including 160 perturbations and a control treatment.

**So2Sat [17]** contains synthetic aperture radar and multispectral optical image patches from remote sensing satellites. Each image in the dataset has 18 channels, of which eight *Sentinel-1* and 10 *Sentinel-2* channels. The dataset consists of 17 classes, each representing a distinct climate zone. We use the city-split version of the dataset, which includes 352K training images and 24K test images.

### 4.3 Results

Tab. 1 shows that DiChaViT outperforms the state-of-the-art ChannelViT by up to $5.0\%$ points on all three datasets: CHAMMI [14], JUMP-CP [12], and So2Sat [17]. For JUMP-CP and So2Sat, we consider two scenarios: tested on all training channels (denoted as "Full") and tested on a subset of channels (denoted as "Partial"). In the full channels setting, our model shows a $1.5 - 2.5\%$ point improvement compared with other baselines on JUMP-CP and So2Sat. When tested on partial channels, DiChaViT demonstrates its robustness by achieving a $1.5\%$ improvement compared with the baselines. This demonstrates that diversifying feature representations in MCI-ViT models boosts both performance and robustness.

Tab. 2 presents a detailed evaluation of DiChaViT and the best baseline model, ChannelViT, when tested on partial channels of the JUMP-CP dataset (with a total of eight channels). For the partial channel evaluation, we exclude some of the channels that the models were trained on and only test the model on the remaining channels. Then, we calculate the average accuracy across all combinations, *e.g.*, testing on seven channels, as shown in column "7", involves averaging the results of $C_8^7 = 8$

Table 3: **Model ablations of DiChaViT.** Removing any component in DiChaViT has a negative impact on overall performance, with significant decreases observed on the *Partial* setting when DCS is removed. Including all components improves performance across all three datasets.

| | CHAMMI [14] | JUMP-CP [12] | | So2Sat [17] | |
| --- | --- | --- | --- | --- | --- |
| Model | Avg score | Full | Partial | Full | Partial |
| **DiChaViT** | **69.66** | **69.19** | **57.98** | **63.36** | **47.76** |
| w/o CDL | 68.07 | 67.66 | 56.87 | 62.20 | 45.74 |
| w/o TDL | 67.61 | 68.12 | 56.62 | 62.39 | 46.87 |
| w/o DCS | 65.32 | 66.03 | 42.37 | 59.20 | 17.88 |

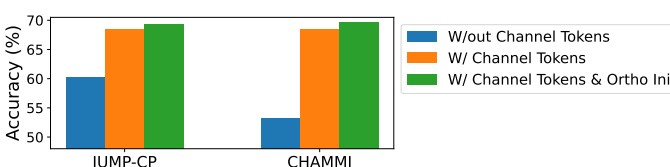

Figure 3: **Performance of DiChaViT** on JUMP-CP and CHAMMI **with** and **without** channel tokens. Using channel tokens with orthogonal initialization (green) improves performance.

combinations (refer to Tab. 9 in the Appendix for detailed results). Our findings consistently show that DiChaViT demonstrates improved robustness when some input channels are missing.

To provide more insight into the contribution of each component of DiChaViT, Tab. 3 presents the model's performance when a component is removed. The results highlight the critical role of the DCS component, as its removal has the most detrimental effect on performance, particularly in the *Partial* setting, with a decrease of 16% and 30% points on JUMP-CP and So2Sat, respectively. The absence of CDL and TDL results in similar performance drops across all datasets. The highest scores are achieved when all components are integrated, indicating that each component plays a crucial role in the model's design. Refer to Tab. 8 in the Appendix for a comprehensive analysis.

### 4.4 Analysis and Discussion

#### 4.4.1 Role of Channel Tokens in MCI-ViT Models

**The role of channel tokens.** In MCI-ViT models such as ChannelViT [18] and ChAda-ViT [15], channel tokens play a crucial role in learning channel-specific features, particularly when dealing with multiple channels where each contains unique information. To assess the impact of channel tokens, we compared the performance of DiChaViT on JUMP-CP and CHAMMI *with* (orange bars) and *without* channel tokens (blue bars), as shown in Fig. 3. The results indicate that DiChaViT demonstrates significant improvements with channel tokens, resulting in 8.0% and 15.0% point increases on JUMP-CP and CHAMMI, respectively, highlighting their importance.

**Orthogonal initialization of channel tokens boosts performance.** As shown in Fig. 3, using orthogonal initialization (green) provides a 1.0% gain on JUMP-CP and CHAMMI. This may suggest that by initializing the weights orthogonally, the model can more effectively capture diverse patterns within the data, resulting in boosting its overall performance.

#### 4.4.2 Ablation on Feature Diversification Losses (CDL and TDL)

**Impact of $\lambda_{\mathrm{CDL}}$ (Eq. 5) in CDL.** Fig. 4(a) and (b) show the performance of DiChaViT (*mean* and *std*) across different values of $\lambda_{\mathrm{CDL}}$ on So2Sat and CHAMMI datasets. We can observe that selecting a value that is too large is not beneficial to the performance. It is worth finding a suitable value for $\lambda_{\mathrm{CDL}}$. On the So2Sat, the best performance is achieved with $\lambda_{\mathrm{CDL}} = 0.001$, while the suitable value for CHAMMI is $0.1$.

**Ablation on TDL (Eq. 4).** Fig. 4(c) reports the performance of our model across different ratios of $\lambda_d$ and $\lambda_s$ in TDL. We set a fixed value of $\lambda_s$ at $0.05$ and vary $\lambda_d$. We observe that using a larger $\lambda_d$ compared with $\lambda_s$ leads to better performance for DiChaViT. This suggests that knowing which channel a token comes from, *i.e.*, the *same* or *different* channel, is necessary. The results indicate

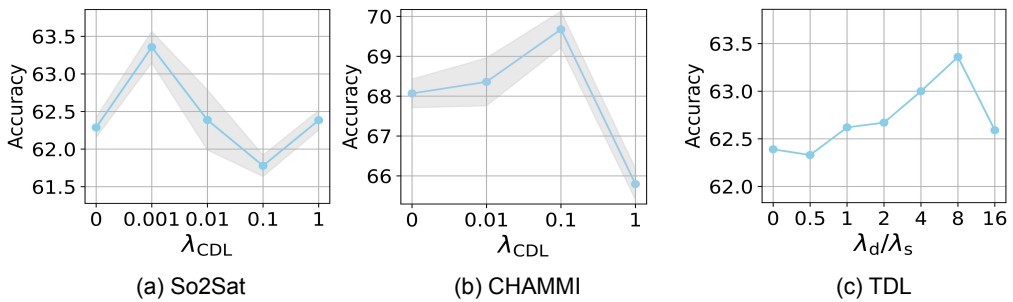

| (a) So2Sat | (b) CHAMMI | (c) TDL |

Figure 4: **Impact of CDL and TDL** on DiChaViT's performance. **(a) & (b)** We demonstrate the average top-1 test accuracy and standard deviation over three runs for different values of $\lambda_{\text{CDL}}$ on So2Sat and CHAMMI. **(c)** Performance with different ratios of $\lambda_d$ and $\lambda_s$ in TDL on So2Sat.

Table 4: **Ablation on the two components of TDL.** *Only $\mathcal{L}_s$* indicates using only within channel tokens (*i.e.*, $\lambda_d = 0$), while *Only $\mathcal{L}_d$* indicates the use of only tokens from different channels in Eq. 4. Incorporating both components in TDL gives the best performance.

|  | So2Sat [17] | CHAMMI [14] |
|---|---|---|
| Only $\mathcal{L}_s$ | 61.43 | 65.47 |
| Only $\mathcal{L}_d$ | 62.50 | 68.15 |
| Both | **63.36** | **69.66** |

Table 5: **Different choices of channel feature $f$ in DCS (Algorithm 1)**. We compare the performance when using the channel tokens ($\text{ch}_i$) and patch tokens (*i.e.*, image patches after passing through the projection layer) to compute the similarity score for sampling.

|  | So2Sat [17] | CHAMMI [14] |
|---|---|---|
| Patch tokens | 63.00 | 65.57 |
| Channel tokens | **63.36** | **69.68** |

Table 6: **Effect of temperature $t_{\text{DCS}}$ on DCS (Algorithm 1).** The first column ($\approx 0$) indicates the use of a very small value of $t_{\text{DCS}}$, which is reduced to selecting the lowest similarity channels. The last column indicates a large value of $t_{\text{DCS}}$, which is reduced to HCS [18]. Using $t_{\text{DCS}} = 0.1$ obtain the best results on So2Sat and CHAMMI datasets.

| Temperature $t_{\text{DCS}}$ | $\approx 0$ | 0.001 | 0.01 | 0.1 | 0.2 | HCS |
|---|---|---|---|---|---|---|
| So2Sat [17] | 62.51 | 63.21 | 63.30 | **63.36** | 61.92 | 62.15 |
| CHAMMI [14] | 67.22 | 66.91 | 68.96 | **69.66** | 66.07 | 66.30 |

imposing stricter constraints on tokens from different channels compared with tokens from the same channel obtains the best performance. Tab. 4 shows the impact of each component in TDL. We see that considering only tokens within the same channels (denoted by "Only $\mathcal{L}_s$") is insufficient, resulting in a significant drop in performance. In contrast, using both $\mathcal{L}_s$ and $\mathcal{L}_d$ in TDL yields the best performance of DiChaViT.

#### 4.4.3 Ablations for Diverse Channel Sampling (DCS)

**Channel feature $f$ in DCS**. Tab. 5 compares the performance of using channel tokens ($\text{ch}_i$) and patch tokens (*i.e.*, image patches after passing through the projection layer) to compute the similarity score for sampling in Algorithm 1 (line 3). We observe that using channel tokens gains better performance on So2Sat and CHAMMI datasets. Note that while channel tokens are shared across all input images, patch tokens differ for each input image.

**Impact of temperature on DCS.** Tab. 6 shows the effect of temperature $t_{\text{DCS}}$ used in Algorithm 1 on DCS. When $t_{\text{DCS}}$ is set to a very small value, as reported in the first column (denoted as "$\approx 0$"), DCS selects channels with the lowest similarity scores to the anchor channel. Conversely, when $t_{\text{DCS}}$ is assigned a large value, denoted as "HCS" in the last column, DCS is reduced to HCS [18], meaning

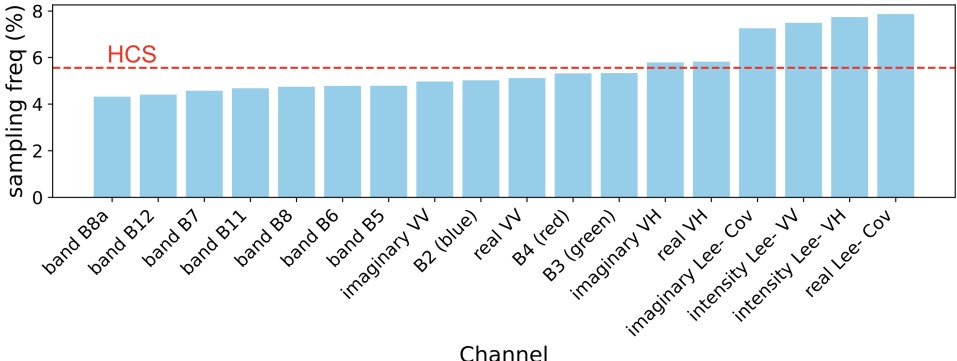

Figure 5: **Comparison of DCS and HCS [18] in terms of the frequency (%) each channel is sampled during training on So2Sat.** Unlike HCS, which provides a uniform distribution for all channels (red dashed line), some channels in DCS are trained much more than others (blue bars). For example, *Real Lee-Cov* channel (rightmost) is sampled twice as much as *Band B8a* (first bar).

that it selects the subset of channels randomly. We find that always selecting the lowest similar channels ($\approx 0$) does not yield the best performance. Instead, setting the temperature to $t_{\mathrm{DCS}} = 0.1$ produces favorable results for both So2Sat and CHAMMI.

**DCS and HCS on the distribution of number sampling of the channels.** Fig. 5 compares the number of times each channel is sampled during training with DCS (blue bars) and HCS [18] (red dashed line). DCS offers a different distribution for its channels compared with HCS, with some channels receiving more training than others. For example, *Real part of Lee-filtered covariance matrix* (*Real Lee-Cov*) in the last bar, is sampled twice as frequently as *Band B8a* channel (first bar).

## 5 Conclusion

In this paper, we present $\mathrm{DiChaViT}$, a model aimed at enhancing feature diversity and robustness in Multi-Channel Imaging (MCI) ViTs. First, we introduce Diverse Channel Sampling, a novel channel sampling strategy that encourages the selection of more distinct channel sets during training, thereby promoting feature diversity. Additionally, $\mathrm{DiChaViT}$ incorporates Token Diversification Loss on the patch tokens and Channel Diversification Loss for channel tokens to further diversify the features learned in MCI-ViTs. Our experiments demonstrate a $1.5 - 5.0\%$ point improvement over state-of-the-art methods on satellite and microscopy imaging datasets. Many of our enhancements are not tied to any specific architecture and can be incorporated into new architectures as they are developed. $\mathrm{DiChaViT}$ represents a promising advancement in addressing the challenges associated with MCI, paving the way for more effective MCI-ViT models.

**Broader Impacts and limitations.** The development of $\mathrm{DiChaViT}$ represents an advancement in MCI, with potential positive impacts such as improved medical diagnosis and accelerated healthcare research. Additionally, its versatility in satellite imaging holds promise for environmental monitoring. However, there are also potential negative impacts, including the risk of bad actors using this research to develop harmful applications, such as invasive surveillance systems. This highlights the importance of ethical considerations and responsible deployment. One of the limitations of our work is that it is not designed to handle novel channels. Generalizing to unseen channels is challenging because it requires establishing a connection between existing and new channels. This is further complicated in the presence of domain shifts, which makes finding the informative channel weights even more difficult. Thus, investigating techniques to adapt to new channels at test time is a promising research direction in MCI. In addition, our approach requires extra hyperparameter tuning, which may necessitate additional compute resources.

## Acknowledgments and Disclosure of Funding

This material is based upon work supported, in part, by the National Science Foundation under award DBI-2134696. Any opinions, findings, and conclusions or recommendations are those of the author(s) and do not necessarily reflect the views of the supporting agencies.

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

# A  Implementation details

We utilize a $\mathrm{ViT}$ small architecture (21M parameters) implemented in $\mathrm{DINOv2}$ [54] as the backbone for all the baselines [5]. Specifically, we use ViT-S/16 (patch size of 16) on CHAMMI and JUMP-CP, and ViT-S/8 (patch size of 8) on So2Sat. The AdamW optimizer [55] is used to train the models, minimizing cross-entropy loss on JUMP-CP and So2Sat, and proxy loss on CHAMMI.

**CHAMMI dataset [14].** The goal of CHAMMI is to train a model to learn the feature representation for the input image. Thus, we use the $[\mathrm{CLS}]$ token at the final layer as the feature representation and train the model to minimize the proxy loss [52]. We then evaluate the model on various tasks following the evaluation code provided by the authors, in which a 1-Nearest Neighbour classifier is used to predict the macro-average F1-score for each task separately [6]. The channel-adaptive interfaces are adapted from the author's implementation code [7]. Besides the model, we incorporate the same data augmentation as introduced by the authors, such as thin-plate-spline (TPS) transformations [56]. We train each model for 60 epochs with a learning rate of 0.00004, and a batch size of 64.

**JUMP-CP [12] and So2Sat [17] datasets**. Following Bao *et al*. [18], the learning rate is warmed up for the initial 10 epochs, peaking at 0.0005 after which it will gradually decay to $10^{-6}$ following a cosine scheduler. We also apply a weight decay of 0.04 to the weight parameters, excluding the bias and normalization terms to mitigate overfitting. Additionally, we use the same data augmentation as used in the code provided by the authors. To get the final prediction, we pass the Transformer encoder's representation for the $[\mathrm{CLS}]$ token into a classifier head to predict the probability of each class. We train each model for 100 epochs, with a batch size of 64 on JUMP-CP, and 128 on So2Sat. We adapt the code provided by the authors [18] for the baselines in our work [8].

**Compute resources.** In this study, experiments were conducted on So2Sat and CHAMMI using a single NVIDIA RTX (48GB RAM) and three Intel(R) Xeon(R) Gold 6226R CPUs @ 2.90GHz. For experiments on JUMP-CP, two NVIDIA RTX A6000 GPUs and six Intel(R) Xeon(R) Gold 6226R CPUs @ 2.90GHz were utilized.

# B  Additional experimental results

**Extended main results**. Tab. 7 shows an extension of the main resulting table in the main paper (Tab. 1), where we include CNN-based ($\mathrm{ConvNeXt}$ backbone [8]) models from [14]. To ensure a fair comparison, we adjust the number of layers in these CNN-based models so that all models in Tab. 7 have approximately 21M parameters. We can observe that in general, $\mathrm{DiChaViT}$ outperforms CNN-based and ViT-based models on the three datasets.

**Extensive ablation results on DiChaViT**. Tab. 8 extends Tab. 3 in the main paper to have a better understanding of the individual effects and contributions of each of the losses. We observe that adding DCS helps improve the performance (*e.g*., by 4% on CHAMMI), and robustness of the model, especially when tested on partial channels (a boost of 35% on So2Sat Partial). Similarly, TDL and CDL also show improvement across the three datasets. For example, TDL improves the performance by 2.5% on CHAMMI and 1.7% on So2Sat on full channels.

**Effect of CDL on channel token distributions.** Fig. 6 illustrates the distributions of channel tokens *with* (blue) and *without* (red) CDL. Each subplot presents the distribution of a trained channel token on the CHAMMI dataset. We observe that CDL results in more flattened distributions with more non-zero values in the channel tokens.

**Attention scores of the** $[\mathrm{CLS}]$ **token to the patch tokens at different layers.** Fig. 7 shows an extended version of Fig. 1(b) in the main paper, where we calculate the attention scores of the $[\mathrm{CLS}]$ token to the patch tokens at layers 4, 8, and 12 (the penultimate layer), and then aggregate them by channel. This indicates that $\mathrm{ChannelViT}$ (top) relies more heavily on specific channels (*e.g*., *microtubules* and *nucleus*) for making predictions, while other channels (*e.g*., *protein* and *er*) are less considered. In contrast, $\mathrm{DiChaViT}$ (bottom) displays more evenly distributed attention scores across channels, indicating that each channel contributes more significantly to the model's predictions.

---

[5]`https://github.com/facebookresearch/dinov2`
[6]`https://github.com/broadinstitute/MorphEm`
[7]`https://github.com/chaudatascience/channel_adaptive_models`
[8]`https://github.com/insitro/ChannelViT`

Table 7: **Test accuracy of channel-adaptive models across multi-channel datasets.** DiChaViT performs better than other CNN- and ViT-based baselines. It shows overall better performance on CHAMMI, especially on Allen and CP, and a $1.5-2.5\%$ improvement on JUMP-CP and So2Sat. "Full" refers to testing on all channels, while "Partial" means testing on a subset of channels. We use *Sentinel-1* channels for So2Sat, and *fluorescence* channels for JUMP-CP.

| Model | Architecture | CHAMMI [14] | | | JUMP-CP [12] | | So2Sat [17] | |
|---|---|---|---|---|---|---|---|---|
| | | Allen | HPA | CP | Full | Partial | Full | Partial |
| HyperNet [42] | ConvNeXt | 58.43 | 65.93 | 26.53 | 53.48 | 10.58 | 58.97 | 41.54 |
| Depthwise [14] | ConvNeXt | 58.76 | 57.60 | 27.39 | 49.34 | 39.88 | 58.60 | 38.87 |
| TemplateMixing [39, 40] | ConvNeXt | 60.21 | 63.44 | 25.98 | 49.74 | 43.74 | 60.79 | 40.61 |
| HyperNet [42] | ViT | 45.17 | 63.90 | 26.23 | 47.07 | 42.43 | 60.73 | 41.88 |
| Depthwise [14] | ViT | 50.35 | **71.52** | 27.74 | 49.86 | 44.98 | 60.41 | 43.41 |
| TemplateMixing [39, 40] | ViT | 49.51 | 64.52 | 25.65 | 52.48 | 43.85 | 55.86 | 37.28 |
| ChAda-ViT [15] | ViT | 67.08 | 60.67 | 24.60 | 65.03 | 42.15 | 56.98 | 12.38 |
| ChannelViT [18] | ViT | 67.66 | 62.14 | 27.62 | 67.51 | 56.49 | 61.03 | 46.16 |
| **DiChaViT** (ours) | ViT | **75.69** | 63.67 | **28.98** | **69.19** | **57.98** | **63.36** | **47.76** |

Table 8: **Extensive Ablation Studies on DiChaViT**. We expanded Tab. 3 in the main paper to show the performance improvements achieved with different combinations of our components, offering more insights into the roles of each component. We report *mean±std* over three runs.

| Exp. | Model | CHAMMI | JUMP-CP | | So2Sat | |
|---|---|---|---|---|---|---|
| | | Avg Score | Full | Partial | Full | Partial |
| 1. | ChannelViT w/o HCS (**ChAda-ViT**) | 63.88±0.34 | 65.03±0.98 | 42.15±2.33 | 56.98±0.46 | 12.38±2.03 |
| 2. | + HCS (**ChannelViT**) | 64.90±0.75 | 67.51±0.35 | 56.49±0.53 | 61.03±0.17 | 46.16±0.40 |
| 3. | + DCS | 67.74±0.33 | 67.90±0.37 | 56.61±0.43 | 62.17±0.23 | 47.30±0.43 |
| 4. | + TDL | 66.27±0.38 | 65.77±0.58 | 43.89±1.89 | 58.68±0.53 | 15.63±5.01 |
| 5. | + CDL | 64.24±0.54 | 66.75±0.57 | 42.74±1.74 | 57.70±0.11 | 15.08±4.00 |
| 6. | + DCS + TDL | 68.07±0.44 | 67.66±0.28 | 56.87±0.78 | 62.20±0.18 | 45.74±0.42 |
| 7. | + TDL + CDL | 65.32±0.48 | 66.03±0.39 | 42.37±1.16 | 59.20±0.43 | 17.88±3.14 |
| 8. | + DCS + CDL | 67.61±0.44 | 68.12±0.60 | 56.62±0.78 | 62.39±0.13 | 46.87±0.24 |
| 9. | + TDL + CDL + HCS | 67.46±0.39 | 67.50±0.90 | 57.10±0.96 | 62.05±0.09 | 45.08±0.60 |
| 10. | + TDL + CDL + DCS (**DiChaViT**) | **69.66±0.43** | **69.19±0.47** | **57.98±0.41** | **63.36±0.11** | **47.76±0.23** |

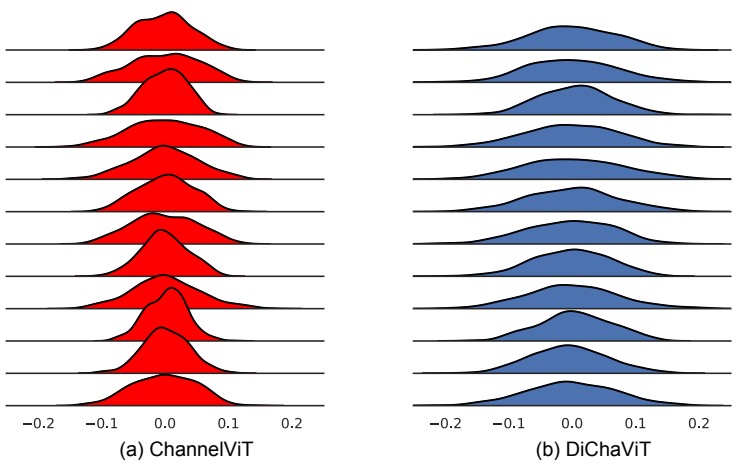

Figure 6: **The effect of Channel Diversification Loss (CDL) on channel embedding distributions**. Each subplot shows the distributions of a channel token after training on the CHAMMI dataset. **(a)** ChannelViT's features (red) are more concentrated around 0. **(b)** In contrast, DiChaViT shows more flattened distributions with more non-zero values (blue).

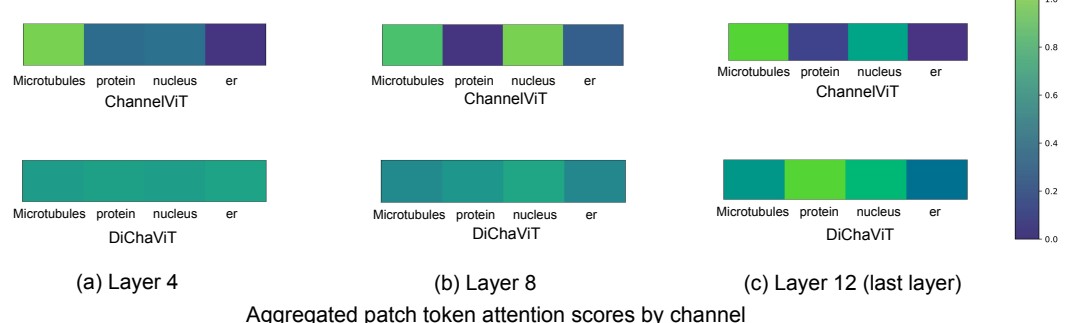

Figure 7: **Attention scores of HPA channels (CHAMMI) across different layers**. We compute attention scores of the [CLS] token to the patch tokens in a layer (layers 4, 8, and 12) and aggregate them by channel. ChannelViT (top) relies on certain channels (*e.g.*, *microtubules* and *nucleus*) to make predictions and less on other channels (*e.g.*, *protein* and *er*). In contrast, DiChaViT demonstrates more evenly distributed attention scores across channels, suggesting that each channel contributes more to the model's predictions.

Table 9: **Detailed performances of ChannelViT and DiChaViT on JUMP-CP in the *leave-one-channel-out at test time* setting**. We present the details of column "7" in Tab. 2 of the main paper. DiChaViT achieves $1 - 2\%$ better performance on each combination compared with ChannelViT.

| Channels at inference | ChannelViT | **DiChaViT (ours)** |
|---|---|---|
| $\{0, 1, 2, 3, 4, 5, 6\}$ | $67.37 \pm 0.60$ | $\mathbf{69.21 \pm 0.19}$ |
| $\{0, 1, 2, 3, 4, 5, 7\}$ | $67.20 \pm 0.59$ | $\mathbf{69.06 \pm 0.20}$ |
| $\{0, 1, 2, 3, 4, 6, 7\}$ | $67.28 \pm 0.53$ | $\mathbf{69.12 \pm 0.16}$ |
| $\{0, 1, 2, 3, 5, 6, 7\}$ | $58.52 \pm 0.63$ | $\mathbf{59.61 \pm 0.17}$ |
| $\{0, 1, 2, 4, 5, 6, 7\}$ | $37.70 \pm 0.60$ | $\mathbf{38.83 \pm 0.46}$ |
| $\{0, 1, 3, 4, 5, 6, 7\}$ | $61.90 \pm 0.48$ | $\mathbf{63.28 \pm 0.31}$ |
| $\{0, 2, 3, 4, 5, 6, 7\}$ | $61.21 \pm 0.41$ | $\mathbf{62.72 \pm 0.28}$ |
| $\{1, 2, 3, 4, 5, 6, 7\}$ | $61.72 \pm 0.48$ | $\mathbf{63.48 \pm 0.20}$ |

**Leave-one-channel-out at test time.** In Tab. 9, we provide individual channel combination results when using seven channels (of eight) of the JUMP-CP dataset for inference. This corresponds to the details in column "7" from Tab. 2 in the main paper, representing $C_8^7 = 8$ different channel combinations. For each combination, we report the *mean* and *std* of the models computed over three runs. Our results demonstrate that DiChaViT gets $1 - 2\%$ better performance for each combination while also providing more stable results (*i.e.*, smaller model variance) than baseline ChannelViT.

