# OpenReview forum: "Enhancing Feature Diversity Boosts Channel-Adaptive Vision Transformers"
_NeurIPS.cc/2024/Conference — NeurIPS 2024 poster_

### Official Review · Reviewer_K1xF · 2024-07-12

**Soundness:** 2
**Presentation:** 3
**Contribution:** 3
**Rating:** 5
**Confidence:** 4

**Summary:**

This manuscript addresses the challenge of learning robust feature representations from datasets with heterogeneous channels, where the number and type of channels vary during training and testing. The authors propose a new DiChaViT, a model for Multi-Channel Imaging (MCI) based on the Vision Transformer (ViT) backbone.

DiChaViT is proposed to improve existing MCI-ViT models like ChannelViT (*Channel Vision Transformers: An Image Is Worth 1 x 16 x 16 Words, Bao et al., 2023*). Indeed, it re-elaborates ideas such as channel-wise patches and channel sampling, which contribute to the model's robustness in handling partial channel configurations at test time. Specifically, DiChaViT addresses previous models' tendency to learn redundant features across channels and patches, limiting their performance and generalizability.

DiChaViT introduces three novelties to promote feature diversity and improve generalization across different channel configurations:

1. **Channel Diversification Loss (CDL):** An additional loss term that encourages learning of distinct channel tokens by maximizing their Euclidean distance.
2. **Token Diversification Loss (TDL):** An additional loss term that aims to diversify the features extracted from each patch token by promoting orthogonality between them (especially between patches belonging to different channels).
3. **Diverse Channel Sampling (DCS):** A novel channel sampling strategy that selects channels based on their dissimilarity.

Overall, they enhance the diversity of learned features, both between different channels and within individual tokens, situating the work in the broader context of learning disentangled representations.

These novel methods, combined with an MCI-ViT model, improve the classification accuracy on multi-channel fluorescence microscopy and satellite image data by 2-5 percentage points, making DiChaViT SOTA on this specific task.

**Strengths:**

The methods proposed are flexible and generalizable. They work independently of the channel configurations and have proven effective in different data domains (microscopy and satellite images). Moreover, they are also agnostic on the specific ViT architecture as they act at the input level. All three proposed novelties outline solid yet intuitive and elegant mathematical formulations.

In particular, Diversity Channel Sampling (DCS) provides a tangible improvement over the existing Channel Sampling approaches often used in MCI-ViT models. The proposed idea is also simple and effective, enabling content-aware channel sampling.

Overall, the novel method proposed by the paper enhances the performance in classifying MC images (with a variable number of channels), making DiChaViT SOTA on this specific task.

**Weaknesses:**

- Despite improving classification accuracy over previous SOTA methods, the paper lacks consistent experiments (or even hypotheses) demonstrating how and why the introduced novelties lead to such performance (e.g., in-depth analysis on the effect of losses on the distribution and content of channel and patch tokens).  In this context, the analyses reported in Appendix B are a good starting point, but the current analyses are too superficial and do not provide enough insights.

- Comparing the accuracy of ChannelViT and DiChaViT for different subsets of channels (as shown in Table 2) is an interesting way to assess the robustness of the model to different channel configurations. However, the conclusion that “DiChaViT consistently demonstrates improved robustness when some input channels are missing” is rather weak and probably not statistically significant due to the large standard deviations. Similarly, the ablation study is extensive and clear, but the assertions about the positive impact of CDL and TDL (see Fig. 4) are not backed up by sufficient statistical evidence. Indeed,
    1. the number of runs is not sufficient to produce reliable uncertainty estimates;
    2. the differences in performance for varying hyperparameters are not remarkable.
Finally, it could be useful to introduce more metrics than simple accuracy.
For these reasons, I would suggest readjusting the emphasis of the claims to what can comortably be claimed.

- There are some points in which the notation and naming is quite misleading:
    1. f() is used both at line 141 to identify a map from a channel to its anchor, and at line 166 to identify a map from a patch to its corresponding channel.
    2. In the CDL paragraph, the authors used c_i to identify the channel token, while in the DCS paragraph c_i is an alias for the channel, and f_i is instead the channel token.
    3. In Fig.2 patch embeddings are referred to as p_i, which is the same notation used for the input patches in the paragraph about TDL.
    4. In algorithm 1 mirror, I would not call the first sampled channel as c_n, since it seems the last channel sampled over a total of n. Something like c_k or $\bar c_i$ would be more coherent.
    5. The name “channel tokens” for the vectors c_i that are concatenated to the patch tokens is a bit unfortunate. Indeed, as far as I am concerned, their role is closer to the one of “patch embeddings” p_i. As a result, I believe that referring to them as “channel embedding” would be a better choice.

**Questions:**

1. Regarding CDL:
    - The orthogonal initialization of channel tokens introduces a strong prior/bias, especially considering that, in MCI, neighboring channels may contain very similar information. Have you tried to see what is the impact of imposing this initialization on tokens belonging to very similar channels? How do you expect the resulting tokens to be once the model is trained?
    - Why did you use the Euclidean distance in the loss? Have you also tried other distances that are more appropriate for multi-dimensional objects?
    - How are the anchors initialized? I believe it would be nice to briefly mention it since it is a key aspect of understanding CDL’s behavior.

2. Why is cosine similarity between channel tokens used in DCS, whereas in CDL euclidean distance is employed instead? I could see potential reasons for that, but I would appreciate it if the authors included a brief note about that in the manuscript.

3. In the first paragraph of section 4.4. the role of channel tokens is analyzed. In this context, why is the evaluation performed on ChannelViT instead of DiChaViT? Indeed, although CDL and the DCS are not applicable without channel tokens, TDL could still be used.

4. The methods seem very flexible to different kinds of data and channel configurations. Have you tried to measure their effectiveness on hyperspectral images with several tens of channels?

**Limitations:**

- I expect the proposed losses CDL and TDL to struggle in the specific case of MCI data where groups of channels share similar semantic information. An example is spectral imaging, where neighboring (wavelength) bands often contain similar information.
In this case, I believe the following:
    1. Regarding the Token Diversity Loss, the choice of introducing a larger penalty for the similarity between patches of different channels is counterintuitive. Indeed, this choice implicitly assumes that the channel information is more important than the spatial one. However, in the context of such images, I would argue that the opposite is instead true in most situations. Likely, two patches referring to the same spatial location but belonging to different channels usually exhibit more similar semantic features than two patches in different locations for the same channel.
    2. Moreover, as already mentioned above, the approach proposed with CDL channel tokens introduces quite a strong prior on the channel token structure, which can be limiting for the datasets mentioned here.
Overall, I see these methods as effective only on MC datasets in which channels contain sufficiently diverse information (but I’m happy to read more results and be proven wrong).

- It would be interesting and meaningful to compare the learned features at different levels of the model (e.g., by visualizing the attention maps) to explicitly assess the disentangling effect brought by these methods.

---

> ### Author Rebuttal · Authors · 2024-08-07
>
> > Can you provide an in-depth analysis on the effect of losses on the distribution and content of channel and patch tokens?
>
> Thank you for your valuable suggestions! Please refer to Fig. 1a for the effect on channel token content and Fig. 7 the effect on channel token distribution.  Additionally, the first two questions of the global response we further analyze model performance, and Fig. 1 of the rebuttal PDF reports the effect on patch tokens. In summary, our contributions result in less redundant information being captured, boosting performance across the board. Additional discussion in the global response.
>
> > Are the gains in Tab. 2 significant (they have large standard deviations)?
>
> We believe that this comment stems from a misunderstanding of what the standard deviations pertain to in Tab. 2.  See Response 2 to Reviewer *gjzo*.
>
> > Are the gains of CDL and TDL significant?
>
> Tab. 1 of the rebuttal PDF shows when including CDL in our full model (exp. 6 & 10) we boost performance by 1-2% with standard deviations from model variance of 0.2-0.4 in most settings, clearly demonstrating significant gains. The gains from TDL are similar (exp. 8 & 10).
>
> > The differences in performance for varying hyperparameters are not remarkable.
>
> This is a benefit of our model as it helps show our model does not need extensive tuning when applied to new datasets.
>
> > Can you use metrics other than accuracy?
>
> Classification accuracy (So2Sat and Jump-CP) and F1 score (CHAMMI) were selected by prior work due to dataset restrictions [14,18].  E.g., CHAMMI’s tasks reveal differences and similarities among cellular phenotypes with a carefully designed evaluation protocol.  Changing this protocol to include other metrics would require collecting a new dataset.
>
> > Notation clarifications/suggestions
>
> We thank the reviewer for carefully reading our manuscript and providing useful suggestions! We will revise our manuscript to make these points clear.
>
> > 1. Regarding CDL: Have you tried to see what is the impact of imposing orthogonal initialization on tokens belonging to very similar channels?
> > Overall, I see these methods as effective only on MC datasets in which channels contain sufficiently diverse information
>
> Our datasets do contain very similar channels, yet models still report improved performance when using orthogonal initialization. E.g., So2Sat (18 channels) contains Red, Green, and Blue channels, which are known to be similar, and even show stronger correlation among Bands B6 - B8a (Fig. 2 of the rebuttal PDF). Yet, we still obtain improvements with CDL (see Fig. 3 main paper, and Tab. 1 of the rebuttal PDF).  We find that some of these similar channels do result in similar tokens in a trained model, i.e., if there is important information the model can overcome an orthogonal initialization.  Additionally, if two channels are similar, then the likelihood they end up learning the same features is high, especially if they are informative features.  Thus, a model may miss learning any unique information contained in only a single channel (resulting in high mutual information in Fig. 1a of the main paper).  As such, in the case of similar channels, it is even more important to encourage diversification so that more than just the redundant information is learned.
>
> > Regarding CDL: Why did you use the Euclidean distance in the loss?
>
> Eq. 1 and L143 note that we use L2-norm + Euclidean distance following ProxyNCA++ [52]. This combination is equivalent to common alternatives like cosine similarity (e.g., as used in [A]). That said, we are happy to compare to other alternatives the reviewer would suggest, but also argue these implementation details do not significantly affect any conclusions we make.
> [A]  Learning transferable visual models from natural language supervision. Radford et al., ICML, 2021.
>
> > How are the anchors initialized?
>
> The trainable channel anchors are also orthogonally initialized, encouraging diversity.
>
> > 2. Why is cosine similarity between channel tokens used in DCS, whereas in CDL euclidean distance is employed instead?
>
> In CDL, we utilize the Euclidean distance with L2-Norm, as mentioned in Line 143, to be consistent with [52]. However, this is equivalent to using Cosine similarity as noted earlier. We will revise our manuscript to clarify this point.
>
> > 3. Can you use DiChaViT instead of ChannelViT when analyzing the role of channel tokens in section 4.4?
>
> We conducted the same experiments using DiChaViT and presented the results in Table 2 of the rebuttal PDF. The conclusion remains the same as ChannelViT’s.
>
> > 4. Have you tried to measure their effectiveness on hyperspectral images with several tens of channels?
>
> We combined the evaluation protocols used by prior work on MCI [14,18], resulting in a more comprehensive study than prior work. That said, we would be happy to explore additional suitable datasets suggested by the reviewer.
>
> > Don’t two patches referring to the same spatial location from different channels exhibit similar semantic features?
>
> We found empirically that a larger penalty for patches from different channels compared to patches within the same channel improves performance. However, in Eq. 4  $\lambda_s$ and $\lambda_d$ balance the loss of tokens belonging to the same channels (first term) and different channels (second term). Thus, one can adjust these hyperparameters according to the dataset if needed.  That said, our experiments already show that similar channels can still benefit.
>
> > It would be interesting and meaningful to compare the learned features at different levels of the model (e.g., by visualizing the attention maps) to explicitly assess the disentangling effect brought by these methods.
>
> We compare the learned features at different layers by visualizing the attention maps in Fig.1 of the rebuttal PDF.  We found that DiChaViT has more evenly distributed attention scores across channels, indicating each channel contributes more to the prediction.

---

> ### Author Response · Authors · 2024-08-10
>
> Hello Reviewer K1xF,
>
> As we are halfway through the discussion period, we are hoping you would read through our rebuttal and let us know if you have any additional questions. We would be happy to answer them, thank you!
>
> -Authors of paper 13979

---

### Official Review · Reviewer_4kDP · 2024-07-13

**Soundness:** 4
**Presentation:** 4
**Contribution:** 3
**Rating:** 7
**Confidence:** 4

**Summary:**

The authors provided the plug-and-play framework for MCI analysis, considering the classification as a downstream task. To do so, they introduced two-channel and token diversification strategies, and in addition, they proposed a new channel sampling strategy for converging the training faster.

**Strengths:**

- Fair and complete ablation studies
- The manuscript is well-written, constructed, and interesting.

**Weaknesses:**

- The authors repeatedly stress that each modification in DCS and TDL helps ensure the model's robustness. What ablation study did you apply to endorses it? Did you do any statistical testing on your experiments?

- The method, as stated, is plug-and-play. Did the authors apply their strategy to other efficient transformer structures with token sampling strategies [1], window-based [2], or linear attentions [3,4]? Because such methods don't have such redundancy exist in the standard Transformer.

The paper needs proofreading. As instances:
- In line 37, the authors used “to” two times repeatedly. Omit one.
- In line 120, it is better to say “as noted in Section 1” rather than “as noted in the Introduction”.
- When citing a work, cite it after the author's name. For example, in line 98, reformat it as “ Recently, Bao et al. [18] …”.

[1] Fayyaz, Mohsen, et al. "Adaptive token sampling for efficient vision transformers." European Conference on Computer Vision. Cham: Springer Nature Switzerland, 2022.

[2] Liu, Ze, et al. "Swin transformer: Hierarchical vision transformer using shifted windows." Proceedings of the IEEE/CVF international conference on computer vision. 2021.

[3] Ali, Alaaeldin, et al. "Xcit: Cross-covariance image transformers." Advances in neural information processing systems 34 (2021): 20014-20027.

[4] Shen, Zhuoran, et al. "Efficient attention: Attention with linear complexities." Proceedings of the IEEE/CVF winter conference on applications of computer vision. 2021.

**Questions:**

Please see the Weaknesses.

**Limitations:**

Please see the Weaknesses.

---

> ### Author Rebuttal · Authors · 2024-08-07
>
> > The authors repeatedly stress that each modification in DCS and TDL helps ensure the model's robustness. What ablation study did you apply to endorses it? Did you do any statistical testing on your experiments?
>
> Table 3 of our paper presents a leave-one-out ablation study, highlighting that best performance is obtained by incorporating each component.  We have expanded upon that study in the rebuttal PDF Table 1 (including the standard deviation from model variance), where we clearly demonstrate the benefits of each model component.  Finally, Tables 1 and 2 of our main paper, Table 1 of our rebuttal PDF, and our response to the second question of Reviewer gjzo also discusses experiments using “partial” sets of channels, i.e., evaluating our robustness to missing channels.  Note that we do not report a separate “partial” result for CHAMMI, as the different image sources within this dataset only contain “partial” sets of channels (so no “full” channel results are possible).
>
>
>
>
> > The method, as stated, is plug-and-play. Did the authors apply their strategy to other efficient transformer structures with token sampling strategies [1], window-based [2], or linear attentions [3,4]? Because such methods don't have such redundancy exist in the standard Transformer.
>
> We appreciate the reviewer’s suggestion to apply our method to efficient transformer structures. Our method can be easily incorporated as a plug-and-play module into various vision transformer models. Following the reviewer's suggestion, we conducted experiments to test our strategy on Adaptive Token Sampler (ATS) [A]. The results, shown in the table below, indicate that using our method on top of ATS results in a 2% performance boost on So2Sat when testing on all training channels. The improvement is especially significant when only partial channels are available at test time, with a performance boost of up to 38%. In the case of CHAMMI, the average score sees a 3% improvement when incorporating our method.
>
> |				| So2Sat Full	| 	So2Sat Partial |  	CHAMMI	|
> |----------------------------------------|--------------------|------------------------------|-------------------------------|
> | ATS				| 58.46±0.59		| 	9.30±1.89 | 	63.02±0.43		|
> |**ATS + (TDL + CDL + DCS)**| **61.52±0.19**	| 	**45.34±0.27**	| **66.18±0.42**	|
>
> [A] Fayyaz, Mohsen, et al. "Adaptive token sampling for efficient vision transformers." European Conference on Computer Vision. Cham: Springer Nature Switzerland, 2022.
>
>
>
>
>
>
> > The paper needs proofreading. As instances:
> In line 37, the authors used “to” two times repeatedly. Omit one.
> In line 120, it is better to say “as noted in Section 1” rather than “as noted in the Introduction”.
> When citing a work, cite it after the author's name. For example, in line 98, reformat it as “ Recently, Bao et al. [18] …”.
>
> We thank the reviewer for carefully reading our manuscript and providing suggestions to enhance the clarity of our paper! We will make the necessary revisions to address these points.

---

> ### Author Response · Authors · 2024-08-10
>
> Hello Reviewer 4kDP,
>
> As we are halfway through the discussion period, we are hoping you would read through our rebuttal and let us know if you have any additional questions. We would be happy to answer them, thank you!
>
> -Authors of paper 13979

---

> > ### Comment · Reviewer_4kDP · 2024-08-12
> > **Official Comment by Reviewer 4kDP**
> >
> > Thank you for providing clarifications. The authors have addressed most of the requested clarifications and included relevant experiments to address other reviewers' comments. Given these improvements, I will maintain my good score. Please add the discussed experiments to the final version of the paper.

---

> > > ### Author Response · Authors · 2024-08-12
> > > **Thank you for your response!**
> > >
> > > Thank you for your positive feedback and for acknowledging the improvements we've made to our work! We appreciate your support and will make sure these discussed experiments are included in the final version.
> > >
> > > -Authors of paper 13979

---

### Official Review · Reviewer_gjzo · 2024-07-13

**Soundness:** 3
**Presentation:** 3
**Contribution:** 2
**Rating:** 6
**Confidence:** 5

**Summary:**

In this paper, the authors present an improved methodology for modeling hyper-spectral multi channel images that can support a variety of channel configurations at test time. The authors reason that the current baselines treat all channels equally and do not consider the diverse qualities of each channel type. Hence they introduce 3 modifications to the training procedure that improves the performance of multi-channel image evaluations over state-of-the-art baseline methods.

The key contributions of the paper include:
1. Orthogonal initialization of Channel Embeddings and Channel Diversification Loss $L_{CDL}$
2. Token Diversification Loss $L_{TDL}$ with different weighting for tokens from the same channel ($\lambda_s$) and different channels ($\lambda_d$).
3. Diverse Channel Sampling strategy that samples diverse set of channels during training

Using the above modification to loss function and training procedure, the authors demonstrate an improvement of 1.5-5% in classification accuracy over the best performing baseline methodology (ChannelViT) on microscopy benchmarks CHAMMI and JUMP-CP and a satellite imaging benchmark So2Sat. The authors have performed further ablation studies to compare the usefulness and contribution of the loss terms.

**Strengths:**

Strengths of the Paper:
* The overall motivation and idea of studying the impact of diversity of channel representations, patch tokens and sampled channels on learned representations and classification performance is interesting. The observations and contributions from the paper will be useful for further exploration and research in this direction.
* The paper is written with sufficient details and clarity of the contributions.
* Well presented introduction and motivation to the problem with sufficient discussion of relevant methods and baselines.
* The authors have provided all the details of their methodology, training parameters, settings and hardware details for reproducibility. They've also agreed to make the code available.
* The authors have chosen relevant strong baselines for comparison and show improvement in classification performance over the baselines with their methodology.
* The authors have performed ablation studies over the new loss terms introduced and demonstrate the performance improvements with each of those loss terms.

**Weaknesses:**

While the overall methodology is well presented and has reasonable intuitive motivation, there are some weaknesses to the proposed methodology.

1. One of the key problems addressed by the channel adaptive image models are generalization to settings where only a subset of channels available. This requires the models to learn both redundant and non-redundant information across channels for generalization to settings where some of the channels are missing or unavailable. While, the authors show channel diversification improving performance of the model, it is important to investigate and interpret where the new setup is improving the performance. Additional experiments on interpretations and comparisons of predictions between baseline methods and proposed methodology might help address this weakness.

2. In Fig 5, the authors compare the distributions of channels sampled during training between HCS (hierarchical channel sampling) and DCS (diverse channel sampling) methodologies. As that DCS samples some channels more compared to other during training, it is important to compare the performance between the baselines and proposed methodologies on those individual channel combination settings. Given the observation above and performance improvements in Table 2 being quite small with large standard deviation, the data is insufficient to the readers to demonstrate strict performance gains.

3. In Table 3, the authors have performed ablation studies by leaving out each of the losses or sampling methods. However, it is also important to show performance improvements with inclusion of each of the losses independently to understand the independent effects and contributions of each of the modifications

**Questions:**

* Can the authors provide a more intuitive explanation for why patch token orthogonality (especially within channel patch token orthogonality) is necessary or might be useful? As the patch tokens are generated using the same set of weights for all patches of images, an identical patch of pixels would produce an identical set of tokens. In such a case, patch token diversification would be unreasonable. The results from Fig 4c further indicates that imposing weaker constraints on tokens from same channels compared with tokens from different channels obtains best performance. So, how is the patch token diversification loss on same channels helpful?

**Limitations:**

* The authors have shown results on limited datasets with smaller dataset sizes. The general applicability of this methodology to large data settings is unknown and would require further experimentation.

---

> ### Author Rebuttal · Authors · 2024-08-07
>
> > 1. It is important to investigate and interpret where the new setup is improving the performance. Additional experiments on interpretations and comparisons of predictions between baseline methods and proposed methodology might help address this weakness.
>
> Thank you for your valuable suggestions! To investigate and interpret where the new setup is improving the performance, we conduct additional analysis in responses 1, 2 and 3 of the Global Response, and Fig. 1&2 in the rebuttal PDF. Additionally, Fig. 7 in the supplementary also gives some insight on the distribution of channel tokens.
>
> > 2.  Given the observation above and performance improvements in Table 2 being quite small with large standard deviation, the data is insufficient to the readers to demonstrate strict performance gains.
>
> Thank you for the question.  We believe that this comment stems from a misunderstanding of what the standard deviations pertain to in Tab. 2.  Specifically, these refer to variances due to the presence or absence of a channel during inference (not model variance). The relatively large standard deviations show some channels are very informative, and removing them results in large drops for both models, but they *do not* suggest our gains over ChannelViT are not significant.
> To illustrate, below we provide individual channel combination results when using 7 channels (of 8), i.e., the “7” column from Table 2, representing 7C8 = 8 channel combinations.  However, for each combination we report the mean and standard deviation of the models computed over 3 runs.  Our results demonstrate that our approach gets 1-2% better performance for each combination while also providing more stable results (i.e., smaller model variance) than ChannelViT.  Other experiments in Tab. 2 reported similar behavior, clearly demonstrating our benefits over prior work.
>
> | Channels at Inference        | ChannelViT           | **DiChaViT (ours)**       |
> |---|----|----|
> | [0, 1, 2, 3, 4, 5, 6]        | 67.37±0.60           | **69.21±0.19**     |
> | [0, 1, 2, 3, 4, 5, 7]        | 67.2±0.59            | **69.06±0.20**     |
> | [0, 1, 2, 3, 4, 6, 7]        | 67.28±0.53           | **69.12±0.16**     |
> | [0, 1, 2, 3, 5, 6, 7]        | 58.52±0.63           | **59.61±0.17**     |
> | [0, 1, 2, 4, 5, 6, 7]        | 37.7±0.60            | **38.81±0.46**     |
> | [0, 1, 3, 4, 5, 6, 7]        | 61.9±0.48            | **63.28±0.31**     |
> | [0, 2, 3, 4, 5, 6, 7]        | 61.21±0.41           | **62.72±0.28**     |
> | [1, 2, 3, 4, 5, 6, 7]        | 61.72±0.48           | **63.48±0.20**     |
>
> > 3. Table 3: It is also important to show performance improvements with inclusion of each of the losses independently to understand the independent effects and contributions of each of the modifications
>
> We thank the reviewer for their suggestions! In Table 1 of the rebuttal PDF, we extended Table 3 in the main paper to have a better understanding of the individual effects and contributions of each of the losses. We observe that adding DCS helps improve the performance (e.g., by ~4% on CHAMMI), and robustness of the model, especially when tested on partial channels (a boost of ~35% on So2Sat Partial). Similarly, TDL and CDL also show improvement across the three datasets. For example, TDL improves the performance by 2.5% on CHAMMI and 2% on So2Sat on full channels.
>
> >  Can the authors provide a more intuitive explanation for why patch token orthogonality (especially within channel patch token orthogonality) is necessary or might be useful?
>
> Patch token orthogonality encourages the tokens generated from different image patches are distinct from each other, helping in preserving and enhancing the discriminative features of the patches. Patches within the same channel can still contain significant variations due to imaging different image regions, and our model helps encourage that they encode different features, if possible.
>
> > An identical patch of pixels would produce an identical set of tokens. Is patch token diversification unreasonable?
>
> In this case the model can not further optimize the diversification loss (it’s at the minimum). That said, like any regularizer, it encourages a bias whose contribution must be carefully tuned.
>
> > The results from Fig 4c further indicates that imposing weaker constraints on tokens from same channels compared with tokens from different channels obtains best performance. So, how is the patch token diversification loss on same channels helpful?
>
> Fig. 4C implies that one may want to apply a larger penalty for the similarity between patches from different channels, compared to within channel. That said, patch token diversification loss on same channel is still helpful, as shown in Table 4 of the main paper.
> Patches within the same channel can still contain significant variations due to different image regions. By enforcing some level of diversification within patches from the same channel, the model can capture finer details and nuances that might be missed.
>
> > Limitations. Experiments are conducted with limited datasets with smaller dataset sizes.
>
> We appreciate the reviewer's feedback. We begin by noting that our experiments are already more extensive than prior work in MCI, as we combine the benchmarks used by two different recent MCI papers [14,18].  In addition, these are similar sizes to many widely used benchmarks (e.g., COCO has 330K images while CHAMMI, JUMP-CP, and So2Sat contain 220K, 217K, and 376K images, respectively).  While they are not suitable for large-scale pretraining, many (if not most) of the applications in MCI simply cannot do so due to lack of data availability and the expertise required to collect new data.  For example, collecting new data and annotations for CHAMMI requires a biologist to perform a time consuming experiment in a lab. As such, we would argue that our experiments are more representative of the real-world applications of MCI.

---

> > ### Comment · Reviewer_gjzo · 2024-08-13
> >
> > The authors have provided sufficient clarification and additional results requested in their rebuttal. Thank you for performing the additional experiments and providing the clarifications. The paper is technically good with no concerns and a moderate to high-impact paper. Hence I have updated my score to reflect that.

---

> > > ### Author Response · Authors · 2024-08-13
> > > **Thank you for your response!**
> > >
> > > Thank you for your feedback and for considering the additional results and clarifications! We appreciate your acknowledgment of the technical soundness and potential impact of our work.
> > >
> > > -Authors of paper 13979

---

> ### Author Response · Authors · 2024-08-10
>
> Hello Reviewer gjzo,
>
> As we are halfway through the discussion period, we are hoping you would read through our rebuttal and let us know if you have any additional questions. We would be happy to answer them, thank you!
>
> -Authors of paper 13979

---

### Official Review · Reviewer_q1Xa · 2024-07-15

**Soundness:** 3
**Presentation:** 4
**Contribution:** 2
**Rating:** 7
**Confidence:** 3

**Summary:**

Multiple channel imaging (MCI) is widely used in different application domains, ranging from medical image analysis to satellite imagery. Each channel can contain information that is orthogonal compared to other channels, which can be useful in downstream tasks. Obtaining such orthogonal signals from individual channels without extracting repetitive information is challenging. In this work, three contributions are made to improve the channel diversity in vision transformer (ViT) class of models. Firstly, to enhance separation between channel specific tokens a contrastive loss, known as channel diversification loss (CDL) is used. Next, patch-level token diversification loss (TDL) ensures extraction of patch-level features that are not dependent on the channels. Finally, masked-training of sorts across channels is performed using diverse channel sampling (DCS). These components result in a composite loss function that is used to demonstrate competitive performance on three MCI datasets. The proposed diverse channel ViT (DiChaViT) method outperforms relevant baseline methods, yielding considerable performance improvements.

**Strengths:**

* MCI data and meaningfully combining channel information, more so, in the presence of missing channels at inference is a challenging task. The proposed contributions that use CDL, TDL, DCS are well-reasoned, and appear to be useful in mitigating some of the known problems within this literature.

* Experiments are comprehensive, with the proposed method showing competitive performance across all three datasets. In some settings, even yielding up to 5% improvements.

* Ablation studies cover a large combination of settings which help understand the contribution of the key contributions.

* The paper is well-structured, performs a thorough literature review of relevant works, and nicely written.

**Weaknesses:**

* **Unseen channels during training**: One of the key limitations of this work, which is also briefly acknowledged by the authors, is that the method is only focused on dealing with missing channels only at inference. Meaning, all channels have to be seen by the model during training. This might not always be the case. I would like the authors to elaborate more on why this is not considered in this work. I would be happy with an explanation in terms of which of the contributions would breakdown for unseen channels. Just to be clear, I am not seeking additional experiments.

* **Balancing loss function**:The composite loss function has multiple hyperparameters: $\lambda_{CDL}, \lambda_s,\lambda_d$, and temperature, $t$. The authors describe the tuning of these parameters individually with discussions on the ranges without elaborating on how to obtain these for other datasets. For instance, $\lambda_{CDL}$ is set to 0.001 for So2Sat and 0.1 for CHAMMI. How were these obtained? Fig. 4 (a,b) show the sensitivity of test accuracy wrt these parameters. And do these magnitudes mean anything? Why the log-scale?

* **Channel anchors**: The discussion on using channel anchors is missing some key information. This could be because I was not aware of the ProxyNCA++ loss referenced in [52]. How are the channel anchors initialised and then selected for each channel? What would happen if instead of learning these vectors, one used a one-hot encoding to specify the channels? Perhaps some clarification here can be useful.

* **Temperature, t**: In L. 148 authors say that results are not sensitive to the choice of $t$ in Eq. 1. However, they set it to a specific (and a peculiar) value of $t=0.07$. And, there is another temperature parameter (?) in Sec. 3.3 when describing DCS which is later reported in Table 5. They are different softmax temperatures, as far as I can see. This is very confusing. And which of the temperatures is more important?

**Questions:**

See points under weaknesses.

**Limitations:**

The authors have addressed the main limitations in the paper.

---

> ### Author Rebuttal · Authors · 2024-08-07
>
> > **Unseen channels during training**: One of the key limitations of this work, which is also briefly acknowledged by the authors, is that the method is only focused on dealing with missing channels only at inference. Meaning, all channels have to be seen by the model during training. This might not always be the case. I would like the authors to elaborate more on why this is not considered in this work. I would be happy with an explanation in terms of which of the contributions would breakdown for unseen channels. Just to be clear, I am not seeking additional experiments.
>
>
> Our contributions should still benefit from a setting requiring inferring unseen channels, but taking advantage of it is not straightforward. A key challenge when generalizing to unseen channels is establishing a connection between existing and new channels.  This is not challenging, as simply identifying a similar channel may not be sufficient.  For example, using the weights for a similar channel may result in simply extracting the same features as that channel, resulting in redundancy that does not improve performance.  Instead, unseen channels require extracting the most informative features, which may not be learned by the most similar channel.  This is further complicated in the presence of domain shifts, which makes finding the move informative channel weights even more difficult.  Thus, this requires a more focused effort to create this mapping that goes beyond the scope of our paper.
>
>
>
> > **Balancing loss function**: The composite loss function has multiple hyperparameters: 𝜆𝐶𝐷𝐿, 𝜆𝑠, 𝜆𝑑, and temperature, 𝑡. The authors describe the tuning of these parameters individually with discussions on the ranges without elaborating on how to obtain these for other datasets. For example, 𝜆𝐶𝐷𝐿 is set to 0.001 for So2Sat and 0.1 for CHAMMI. How were these obtained?
>
> Hyperparameters are set using grid search over a validation set.  We shall update our plot to include both the test and validation results in our revised paper.
>
>
>
>
>
>
> > Fig. 4 (a,b) show the sensitivity of test accuracy wrt these parameters. And do these magnitudes mean anything? Why the log-scale?
>
> The magnitudes show how these parameters affect the model performance. For example, a very high $\lambda_{𝐶𝐷𝐿}$ value could cause the model to excessively prioritize diversifying the channel token features, which may negatively affect the performance on downstream tasks. In contrast, if $\lambda_{𝐶𝐷𝐿}$ is too small, it may not effectively diversify the learned channel token features, leading to underutilization of its benefits.
>
> Similarly, $\lambda_{s}$ and $\lambda_{d}$ control the constraint of tokens belonging to the same channels and different channels in Eq. 4.
>
> Using a log scale allows us to visualize a wide range of parameters to observe the impact of these parameters on the final performance.
>
>
>
>
>
>
> > **Channel anchors**: The discussion on using channel anchors is missing some key information. This could be because I was not aware of the ProxyNCA++ loss referenced in [52]. How are the channel anchors initialised and then selected for each channel?
>
> As noted in L132 of our paper, each channel has its own anchor initialized orthogonally to the other channel anchors.  There is no need for them to be “selected” as they are trained as their anchor. I.e., in So2Sat there are 18 channels, so we would orthogonally initialize 18 channel anchors, each trained to correspond to one specific channel.
>
> > What would happen if instead of learning these vectors, one used a one-hot encoding to specify the channels? Perhaps some clarification here can be useful.
>
> Each channel anchor is the same dimension as the channel token so that we could compute similarities between them.  For the 384-D dimension channel tokens used in our experiments, we would have a 384-D channel anchor.  A one-hot encoding on the 18 channel So2Sat dataset would effectively ignore 384-18=366 of the channel token features (as their loss contribution would always be 0 in a one-hot encoding of the anchors).  Thus, this would, in large part, make our CDL loss completely ineffective in practice.
>
> > **Clarification on two Temperatures**:  In L. 148 authors say that results are not sensitive to the choice of 𝑡 in Eq. 1. And, there is another temperature parameter (?) in Sec. 3.3 when describing DCS which is later reported in Table 5. They are different softmax temperatures, as far as I can see. This is very confusing. And which of the temperatures is more important?
>
> Thank you for carefully reading our paper and spotting out the confusion regarding the temperatures! In line 148, the temperature $t$ controls the sharpness of the probability distribution when using CDL. We observed that the model is not sensitive to the value of $t$, and simply fixing it consistently yields good results across datasets.
>
> The second temperature $t$ mentioned in Section 3.3 (Algorithm 1) is used to control the distribution of CDS. This $t$ is the one presented in Table 5 and is more sensitive in our experiments.
>
> We will revise the manuscript by using $t_{CDL}$ and $t_{CDS}$ for these temperatures to make them clear.
>
>
>
> >  However, they set it to a specific (and a peculiar) value of 𝑡=0.07.
>
> In Eq. 1 of our paper the temperature $t$ is represented as the denominator of a fraction.  We used grid search centered around the denominator value of $t=1/9$ used by ProxyNCA [52].  None of the results were statistically different, but we used $t=1/14=0.07$ as it had the best average result.

---

> > ### Comment · Reviewer_q1Xa · 2024-08-11
> > **Response to author rebuttal**
> >
> > The authors have addressed most of the clarifications sought by me, and also added relevant experiments in addressing other reviewer comments. In particular, the justification for dealing with unseen channels is convincing. I would suggest the authors include this in limitations in the final version of the paper.
> >
> > I will raise my score to Accept.

---

> > > ### Author Response · Authors · 2024-08-11
> > > **Thank you for your response!**
> > >
> > > Thank you for your thorough review and positive feedback on our revisions! We will make sure to highlight the justification for dealing with unseen channels in the limitations section of the final version.

---

> ### Author Response · Authors · 2024-08-10
>
> Hello Reviewer q1Xa,
>
> As we are halfway through the discussion period, we are hoping you would read through our rebuttal and let us know if you have any additional questions. We would be happy to answer them, thank you!
>
> -Authors of paper 13979

---

### Author Rebuttal · Authors · 2024-08-07

We thank the reviewers for their constructive comments and suggestions on the result analysis!
To get more insights, we conducted some additional analyses and attached the figures and tables in the PDF for the rebuttal.

## 1. How Do Channel-Specific Attention Distributions Differ Between ChannelViT and DiChaViT?
To investigate and interpret where the new setup is improving the performance, we look into the attention map of each channel and observe that ChannelViT relies heavily on a subset of channels to make a prediction, while DiChaViT pays attention more evenly across channels.

Fig.1 of the rebuttal PDF, we calculate the attention scores of the [CLS] token to the patch tokens in a layer, such as layers 4, 8, and 12 (last layer), and then aggregate them by channel. This indicates that ChannelViT (top) relies more heavily on specific channels (e.g., *microtubules* and *nucleus*) for making predictions, while other channels (e.g., *protein* and *er*) are less considered. In contrast, DiChaViT (bottom) displays more evenly distributed attention scores across channels, indicating that each channel contributes more significantly to the model's predictions.

## 2. What is the benefit of having evenly distributed attention scores across channels?
To better understand why having more uniform attention scores across channels can enhance robustness and performance, we conducted a test where we removed one channel at test time. We observed that ChannelViT's performance dropped more than DiChaViT's.

For this setting, we trained ChannelViT and DiChaViT on CHAMMI, and then removed the *microtubules* channel during inference on HPA. This means we used only the remaining three channels (*protein*, *nucleus*, and *er*) for testing. As discussed in Section 1 above, ChannelViT relies on this channel to make predictions. As expected, we observed that ChannelViT's performance dropped more compared to DiChaViT (0.38 vs. 0.22), shown in the table below. This suggests that DiChaViT has a better ability to integrate and extract information from all the channels, making it less reliant on any single channel for accurate predictions.


|                    |Full Channels      |  Removing microtubules |   Avg Performance Drop    |
|-------------------|--------------------|------------------------|------------------------|
|ChannelViT         |   62.17±0.10       |   61.79±0.11           |   0.38                 |
|DiChaViT           | 	63.59±0.12      |   63.28±0.13            |   0.22                 |


## 3. Where does DiChaViT improve performance?
We observe that the performance gets improved across classes, such as *Endoplasmic Reticulum* (from 910 to 1070 correct predictions) as shown in the confusion matrices below.

In this setting, we trained ChannelViT and DiChaViT on CHAMMI, and tested the models on 3 novel classes (*Cytosol*, *Endoplasmic Reticulum*, and *Nucleoplasm*).

**ChannelViT’s Confusion Matrix**
|               | Cytosol | Endoplasmic Reticulum | Nucleoplasm |
|---------------|---------|-----------------------|------------|
| **Cytosol**                | 2047    | 962                   | 256         |
| **Endoplasmic Reticulum**  | 1126    | 910                   | 216         |
| **Nucleoplasm**            | 241     | 179                   | 3234        |


**DiChaViT’s Confusion Matrix**
|                         | Cytosol | Endoplasmic Reticulum | Nucleoplasm |
|-------------------------|---------|-----------------------|-------------|
| **Cytosol**             | **2109**    | 848                   | 285         |
| **Endoplasmic Reticulum** | 996     | **1070**                  | 186         |
| **Nucleoplasm**         | 265     | 158                   | **3393**        |



In addition,  their classification reports are shown below. We observe that DiChaViT gains improvement across all metrics (precision, recall, and f1-score) for all classes. This indicates that the model has generally become better when working on novel classes at test time. The most significant improvements are seen at *Endoplasmic Reticulum*.

**ChannelViT’s Classification Report**
| Class                    | Precision | Recall | F1-Score | Support |
|--------------------------|-----------|--------|----------|---------|
| Cytosol                  | 0.60      | 0.38   | 0.47     | 5318    |
| Endoplasmic Reticulum    | 0.44      | 0.24   | 0.31     | 3830    |
| Nucleoplasm              | 0.87      | 0.54   | 0.67     | 6018    |
| **Micro Avg**            | 0.68      | 0.41   | 0.51     | 15166   |
| **Macro Avg**            | 0.64      | 0.39   | 0.48     | 15166   |
| **Weighted Avg**         | 0.67      | 0.41   | 0.51     | 15166   |




**DiChaViT’s Classification Report**
| Class                    | Precision | Recall | F1-Score | Support |
|--------------------------|-----------|--------|----------|---------|
| Cytosol                  | 0.63      | 0.40   | 0.49     | 5318    |
| Endoplasmic Reticulum    | 0.52      | 0.28   | 0.36     | 3830    |
| Nucleoplasm              | 0.88      | 0.56   | **0.69**     | 6018    |
| **Micro Avg**            | 0.71      | 0.43   | **0.54**     | 15166   |
| **Macro Avg**            | 0.67      | 0.41   | **0.51**     | 15166   |
| **Weighted Avg**         | 0.70      | 0.43   | **0.53**     | 15166   |

---

### Decision · Program_Chairs · 2024-09-25

**Decision:**

Accept (poster)

**Comment:**

The paper introduces DiChaViT an enhanced modeling of hyper-spectral images by increasing the diversity of multiple channels using orthogonal channel embedding, token diversification, and diverse channel sampling strategy. The proposed pipeline boosts the classification performance on multi-channel image settings such as satellite and microscopy imaging by encoding MCI into a more robust embedded feature representation. The paper has received all accept reviews. There is consensus among reviewers that the utility of DiChaViT is novel and it is well compared with several baseline approaches and can generalize to different data domains. Overall the paper is well reasoned, motivated and presented which opens a new direction on MCI. Further, the details of the proposed DiChaViT are provided in comprehensive manner and thorough experiments with ablation studies are demonstrated. Some concerns were raised by reviewers on (a) limitation of the proposed method where all channels are needed for training which raised the question about channels that are not seen during training, (b) sensitivity of composite loss hyper-parameters and their transferability to other dataset distribution, (c)  Initialization of channel anchors, (d) sensitivity on temperature parameter, (e) the rational and interpretability of DiChaViT’ design and how it compares with baseline approaches, (f) ablation on each losses individually, (g) intuition behind patch token orthogonality, and (h) notation typos and naming of variables can be miss leading. Authors have addressed most of reviewers concerns during rebuttal. The AC finds strong support from reviewers to merit the paper for publication. It is highly encouraged for the authors to take the advantage on the discussions raised by reviewers as highlighted above for their final revision using from both pre-/post-rebuttal phase comments.